

# Global Cloud Property Models for Real Time Triage Onboard Visible-Shortwave Infrared Spectrometers

Macey W. Sandford[1], David R. Thompson[2], Robert O. Green[2], Brian H. Kahn[2], Raffaele Vitulli[3], Steve Chien[2], Amruta Yelamanchili[2], Winston Olson-Duvall[2]

[1]University of Hawai'i at Manoa, Hawai'i Institute of Geophysics and Planetology, Department of Earth Sciences, Honolulu, HI, USA
[2]Jet Propulsion Laboratory, California Institute of Technology, Pasadena, CA, USA
[3]European Space Agency, European Space Research and Technology Center, Noordwijk, Netherlands

*Correspondence to:* Macey W. Sandford (msandfor@hawaii.edu)

**Abstract.** New methods for optimizing data storage and transmission are required as orbital imaging spectrometers collect ever-larger data volumes due to increases in optical efficiency and resolution. In Earth surface investigations, storage and downlink volumes are the most important bottleneck in the mission's total data yield. Excising cloud-contaminated data onboard, during acquisition, can increase the value of downlinked data and significantly improve the overall science performance of the mission. Threshold-based screening algorithms can operate at the acquisition rate of the instrument but
require accurate and comprehensive predictions of cloud and surface brightness. To date, the community lacks a comprehensive analysis of global data to provide appropriate thresholds for screening clouds or to predict performance. Moreover, prior cloud screening studies have used universal screening criteria that do not account for the unique surface and cloud properties at different locations. To address this gap, we analyzed the Hyperion imaging spectrometer's historical archive of global Earth reflectance data. We selected a diverse subset spanning space (with tropical, midlatitude, arctic, and Antarctic
latitudes), time (2005-2017), and wavelength (400 – 2500 nm) to assure that the distributions of cloud data are representative of all cases. We fit models of cloud reflectance properties gathered from the subset to predict locally and globally applicable thresholds. The distributions relate cloud reflectance properties to various surface types (land, water, and snow) and latitudinal zones. We find that taking location into account can significantly improve the efficiency of onboard cloud screening methods. Models based on this dataset will be used to screen clouds onboard orbital imaging spectrometers, effectively doubling the
volume of usable science data per downlink. Models based on this dataset will be used to screen clouds onboard NASA's forthcoming mission, the Earth Mineral Dust Source InvesTigation (EMIT).

## 1 Introduction

Imaging spectrometers, also known as hyperspectral imagers, collect images in the form of three-dimensional data cubes: a two-dimensional image of the target surface in the field of view and swath of the instrument with a continuous spectrum in the
third dimension. With the recent decommissioning of Hyperion, an early imaging spectrometer onboard NASA's Earth



Observer 1 (EO-1), many space agencies are now planning or operating a new generation of orbital imaging spectrometer missions, e.g. HISUI (Tachikawa et al., 2012), EMIT (Green et al., 2018), EnMAP (Guanter et al., 2015), CHIME, PRISMA, DESIS, TRUTHS. At the time of this writing, NASA is studying dramatically enhanced imaging spectrometer architectures to provide measurements with global coverage (National Academies of Sciences, Engineering, and Medicine 2018). As instrument capabilities grow, the swath width and length of these instruments permits growing coverage areas. Due to the high dimensional nature of these datasets, the duty cycle of these instruments is expected to be limited by data volume. They are operated with a store-and-forward mode (Williams et al., 2002), where the data are stored onboard in a limited "flight recorder" and transmitted when a ground station is within view or when a manual transfer occurs. This limits the bandwidth from the satellite to the ground and ultimately the total data yield of the mission. Consequently, optimizing the downlink from orbital remote sensing spacecraft can increase the science productivity of these missions. One promising strategy to reduce data volumes involves analyzing images onboard as they are being collected, excising contaminated or irrelevant scenes, and preserving good quality data for preferential storage and downlink. (e.g. Thompson et al., 2014; Doggett et al., 2006; and Altinok et al., 2016).

Clouds are the most promising target for onboard screening since they are a common yet unpredictable contaminant that prevents direct observations of surface features. Previous studies indicate that clouds account for over half of the annual sky cover globally (Mercury et al., 2012; Eastman et al., 2011; King et al., 2013; Mace et al., 2009; Rossow and Schiffer, 1999). Thus, onboard cloud screening could approximately double the science productivity per downlink without changing the total stored or transmitted data volumes. Doggett et al. (2006) onboard EO-1 used expert decision tree and support vector machine learning for onboard cloud classification. Altinok et al. (2016) onboard IPEX and Wagstaff et al. (2018) onboard EO-1 used random decision forest machine learning for onboard cloud classification. Though these were not continuous cloud-screening tools, they have determined that cloud screening is in fact a viable solution to the data reduction problem. NASA's EMIT mission has since baselined this capability (Green et al, 2019), and will be the first imaging spectrometer to use continuous onboard cloud screening operationally. These new generation missions will build on a long history of cloud screening algorithms in ground data systems. Most of these screening algorithms treat cloud detection as a classification problem, where attributes of the instrument data are used to determine whether clouds are present. They use a wide variety of techniques including spectroscopically-estimated atmospheric properties (Taylor et al., 2016), band- or regionally-specific threshold tests (Ackerman et al., 1998), spatial variability (Martins et al., 2002), reflectance models of surface and atmosphere (Gómez-Chova et al., 2007), or data-driven machine learning methods (Yhann et al., 1995). Onboard algorithms carry special requirements: they must be simple, for encoding into instrument hardware or low-power spaceflight computers, and they must use only data available to the spacecraft at the time of acquisition. Finally, because the screened data is irrevocably lost, its behavior should be transparent to the operators and tunable to be more aggressive or conservative depending on their error tolerance. In addition, the algorithm's statistical properties and error rates should be well understood.

Most prior screening approaches, including those designed for onboard use, strive for global performance – a cloud classifier that works the same way for all locations globally. However, the nature of the classification problem changes



depending on location. Neither the coverage nor the spectral appearance of clouds is uniform. Clouds cover more of the oceans (68-72%) on Earth annually than land (54-58%), and tropical regions are exceptionally cloudy (Mercury et al., 2012; Eastman et al., 2011; King et al., 2013; Rossow and Schiffer, 1999). Moreover, cloud optical properties vary regionally due to the different processes involved in their formation and evolution (Thompson et al., 2018). Finally, the optimal thresholds for a particular excision scenario also depend on the expected brightness of the surface (Thompson et al., 2014). Nevertheless, prior

studies have relied on universal cloud models and excision criteria, applying the same models for use across the globe. Considering cloud fractions as a function of surface type and latitude could lead to more precise cloud detection. Based on this, we hypothesize that location-specific cloud models can improve performance relative to global methods.

        To test this hypothesis, we refine the cloud-screening algorithm previously introduced in Thompson et al. (2014). The original algorithm uses three bands of interest to distinguish clouds from other surface types (El-Araby et al., 2005; Ackerman

et al., 1998; Williams et al., 2002; Griffin et al., 2003). This has the desired properties of being simple, fast, transparent, and tunable by ground operators. Here, we analyze the Hyperion global archive to provide globally applicable models that are parameterized by latitude and surface type, enabling reflectance thresholds to predict the classification of cloud-contaminated data and non-cloud-contaminated data (land, water and snow) in new scenes for EMIT and other future orbital missions.

## 2 Methods

Most cloud screening algorithms are *classifiers* – they analyze independent datapoints and decide which ones should be flagged as cloudy. These individual datapoints can be pixels within a data cube (Altinok et al., 2016), in which case each location is labeled independently. Other algorithms label segments within the scene (Thompson et al., 2014), or entire image cubes. In this study, our datapoints are individual pixels that are single locations within a scene, or image. Each pixel is associated with a complete measured radiance spectrum and each receives an independent classification as either clear or cloudy. We analyzed

data from the Hyperion imaging spectrometer, an instrument onboard NASA's EO-1 satellite, which collected a globally representative data set over more than a decade (Thompson et al., 2018).

        After calculating TOA reflectance, we labeled each pixel as one of four categories: land, water, snow and clouds. We then accumulated brightness distributions for each pixel type, describing their respective TOA values in each of the following wavelengths of interest: 447.17 nm, 1245.36 nm, and 1648.90 nm (Thompson et al., 2014). Our procedure took the following

steps: we acquired a dataset of representative spectra from a historical data archive; we chose channels or bands that would be used to classify pixels; we calculated pixel brightness distributions for different surface types and latitudes; and finally, we optimized channel thresholds given the distributions and false alarm requirements. These channel thresholds can predict the optimal TOA values for screening clouds in new scenes based on advance knowledge of surface types and viewing geometry.



## 2.1 The Data Set

The Hyperion instrument was a push broom imaging spectrometer that operated onboard NASA's Earth-Observing One (EO-1) spacecraft from 2000-2017. It acquired spectra in 220 channels spanning 357 to 2576 nm at approximately 10 nm spectral resolution. Hyperion operated in a targeted acquisition mode, acquiring images at specific locations of interest. Each map was approximately 7.7 km in width, and typically 42 kilometers in length. During its 17-year operational lifetime, Hyperion acquired tens of thousands of maps across diverse areas including arctic, Antarctic, oceanic, and terrestrial surfaces. We

selected a subset of 102 Hyperion images over the entire time range of the mission for our study. This random set incorporated various sections of latitude to ensure a global representation of measurements. The following subsets were included: Tropics (23.5°S to 23.5°N), Arctic (66.5°N to 90°N), Antarctic (66.5°S to 90°S), and Midlatitudes (66.5°S to 23.5°S and 23.5°N to 66.5°N). Most Hyperion data was acquired over land, so we included a subset of longitudes spanning the Pacific Ocean (121°E to 180°E and 121°W to 180°W) to capture the spectral properties of water. We transformed the radiance measurements to

Top of Atmosphere (TOA) reflectance values to remove the variability caused by solar geometry.

We next extracted a subset of informative bands to use in the cloud classification decision. To be clear, Earth's total TOA energy flux constitutes the total incoming solar radiation, the consequential outgoing *reflected* shortwave radiation from the clouds and surface, and the outgoing *emitted* longwave radiation from Earth's surface, atmosphere, and clouds (e.g., Trenberth et al. 2009). Some other cloud screening algorithms do use longwave channels which provide extra sensitivity to

high altitude clouds (Mercury et al., 2014). However, this study sought to discriminate cloud and clear locations using the Visible-ShortWave Infrared (VSWIR) region alone. This made our results directly applicable to future missions measuring solar-reflected wavelengths. We selected three wavelengths of interest, namely 447.17 nm, 1245.36 nm, and 1648.90 nm, based on previous studies distinguishing clouds from land, water, and snow (Ackerman et al., 1998; Griffin et al., 2003; El-Araby et al., 2005; Thompson et al., 2014). Clouds and snow had a high reflectance in the 447 nm band while land and water

did not. The near-infrared (1245 nm) and shortwave-infrared (1650nm) band reflectance values effectively discriminated between clouds and snow. Snow had a slightly lower reflectance in the 1245 nm band than clouds while the 1650 nm band showed snow as even less reflective than clouds (Griffin et al., 2003). We used just these three channels, since a small subset of bands enables threshold-based algorithms to be encoded easily into instrument electronics hardware, for real-time execution at the native frame rate of the spectrometer.

## 120    2.2 Classification

Accurate ground truth classifications were necessary to define cloud and land statistics for building the classifier. They were also useful for evaluating the resulting model's accuracy by comparing predictions against manual labels. Pixels were hand-labeled to assure accuracy in classifying each surface type (Fig. 1). We then used these data to relate the pixels' categories of land, water, snow, or clouds to their TOA reflectance values. After fitting this model, the manual classification will be used to



verify the model's classification accuracy. We only labeled opaque clouds; all pixels bordering various classification types
      were labeled as "ambiguous" to avoid misclassification.

      After manual labeling, we associated the surface types with the TOA reflectance value in each of the three bands of
      interest. This association produced three-dimensional frequency distributions of TOA reflectance values based on wavelength
      and surface type (Fig. 2). They described the conditional probability of a pixel's TOA reflectance value given its classification

as cloud or clear sky, $c_1$ and $c_2$ respectively ($P(y|c_{1,2})$). The non-cloud distribution contained all non-cloud surface types (land,
      water, and snow). To assess the classification power over each surface, we tracked the brightness distributions of each surface
      type independently. To capture the effect of different climates, we represented the distributions as a function of latitudinal
      zone, $P(y|x,c)$, where x was the zone of interest.

## 2.3 Algorithms

Our cloud-screening approach predicted scene-specific thresholds in three bands for real-time use onboard (Thompson et al.,
      2014). Our cloud-screening algorithm defined an exclusion region $R \subseteq \mathbb{R}^d$, i.e. a range of TOA reflectance values for which a
      pixel was classified as cloudy. In other words, it mapped the pixel brightness values to a binary classification $c = f(\mathbf{y}): \mathbb{R}^d \mapsto$
      $\{c_1, c_2\}$. A vector y represented the spectrum of the pixel being classified. Thus, the decision rule for this classification was,

$$f(\mathbf{y}) = \begin{cases} c_1, \ if \ \mathbf{y} \in R \\ c_2, \ if \ \mathbf{y} \notin R \end{cases},$$  (1)

where R was defined with a set of thresholds, φ (in this case a triplet). Any pixel exceeding all three thresholds simultaneously
      is classified as cloud-contaminated (Fig. 3). We then used the following expected loss function, where $\alpha_{FP}$ and $\alpha_{FN}$ were the
      false positive and negative penalties, respectively,

$$E[\mathcal{L}] = \int_R \alpha_{FP} P(c_1|\mathbf{y}, \mathbf{x}) d\mathbf{y} + \int_{\mathbb{R}^d/R} \alpha_{FN} P(c_2|\mathbf{y}, \mathbf{x}) d\mathbf{y},$$  (2)

      A false positive penalty applied to cases where clear pixels were classified as cloud contaminated. $P(y|c_1)$ and $P(y|c_2)$ were the

probability of encountering a cloud-contaminated pixel and a clear pixel, respectively. $P(c_1)$ and $P(c_2)$ were the prior probability
      of clouds and clear sky, respectively, based on an historical average. To eliminate any bias due to historical observations, an
      "uninformed" prior assigning equal probability to all classes could also be used. Minimizing this function (Eq. 2) produced the
      optimal threshold for the given conditions defined by x. Using Bayes' rule and assuming independence (Thompson et al.,
      2014), the expected loss function could be decomposed into the respective likelihoods and priors for the posterior described

above,

$$E[\mathcal{L}] = \int_R \alpha_{FP} P(\mathbf{y}|\mathbf{x}, \mathbf{c_1}) P(c_1) d\mathbf{y} + \int_{\mathbb{R}^d/R} \alpha_{FN} P(\mathbf{y}|\mathbf{x}, \mathbf{c_2}) P(c_2) d\mathbf{y},$$  (3)

      Thus, we could use the likelihood, or sampling distribution, created from the Hyperion sample set to minimize our expected
      loss and produce predictive thresholds for screening.  As in Thompson et al. (2014), we represented probability distributions
      through histogram counts, creating a 3D table with one dimension for each of the three wavelengths. This allowed a fast-





recursive calculation of equation (3) when searching over thresholds. For a given false positive penalty, we searched over all possible thresholds and selected the one which produced the lowest effective loss.

## 3 Results

The globally representative sample set of imaging spectroscopy data provides a comprehensive sample of TOA reflectance for various surface types in space, time, and wavelength as well as a prediction model for screening cloud-contaminated data

onboard orbital imaging spectrometers. This section discusses our findings concerning excision thresholds, cloud brightness, the potential improvement yield of downlink using the cloud-screening algorithm, a comparison of resulting cloud fractions in our data set with previous literature, justifying the empirical error of our data set, and the implications of our study and cloud-screening tool for future missions.

### 3.1 Cloud Screening Thresholds

The model used for cloud screening was developed using cloud brightness distributions in TOA values as a function of time, space, and wavelength. The brightness distributions collected from the Hyperion sample set represent cloud and non-cloud brightness values in TOA units, for each band (447.17 nm, 1245.36 nm, 1648.90 nm). The sample set scenes cover representative regions across the globe (Table 1).

      The output of the algorithm was a threshold triplet that defined the exclusion region, i.e. the minimum TOA

reflectance values of opaque cloud-contaminated data (discussed in Sect. 2.3). The penalties in the expected loss function determined our tolerance for errors; a higher false positive penalty led to a more conservative threshold and a smaller exclusion region. As we hypothesized, we were able to improve performance further by considering the expected surface properties when defining the exclusion region. For example, since clouds and snow had similar reflective properties in two of the bands used, snow scenes were most challenging and dominated the threshold criterion. Consequently, one could use a more aggressive

threshold triplet to screen clouds outside Arctic regions, improving performance without incurring misclassifications. The result of a conservative threshold ($\alpha_{FP}$=1000) calculation, a moderate threshold ($\alpha_{FP}$=100) calculation, and an aggressive threshold ($\alpha_{FP}$=10) calculation are shown in a two-dimensional histogram representing all scenes in the sample set (Fig. 4).

      The optimal thresholds, where the loss was minimized, at various false positive rates using the Hyperion sample set are shown in Table 2. The thresholds were defined differently for each latitudinal zone. Sect. 3.4 describes the statistical

validation tests we used to ensure the size and breadth of our subset, or sampling distribution, was sufficient to predict these thresholds for future scenes.

### 3.2 Cloud Brightness

The Hyperion global data set was sampled and manually classified to understand cloud brightness as a function of time, space, and wavelength. We collected the TOA values in a 3-dimensional histogram, one axis for each wavelength studied. This



produced a probability distribution of TOA values of clouds globally that we used to predict the classification of TOA values for future scenes. In order to verify if classifying clouds depending on their latitudinal zone would produce a lower false alarm rate, we subset our scenes based on latitude and surface type.

     Table 3 presents the mean TOA reflectance values for each latitudinal zone and wavelength. Cloudy pixels had higher TOA reflectance values while non-cloudy pixels had generally lower TOA values (Fig. 3). We found our data to align with

the general properties of non-cloud surface types discussed in Ackerman et al. (1998). Specifically, pixels with snow had low TOA reflectance values at 1250 nm and even lower TOA reflectance values at 1650 nm, while having high TOA reflectance properties at 447 nm.

     Our results showed that mean cloud TOA brightness differed in the bands studied as a function of latitudinal zone and surface type; Tropics, Arctic, Antarctic, Midlatitudes, and Pacific Ocean areas (Table 4). The starkly higher differences

seen in the Arctic region show that this region needed more conservative thresholds than a global "universal" model which did *not* calculate unique screening thresholds based on latitude. The difference in these mean values indicates that the optimal thresholds assigned for the classification of clouds in each area should also differ (Table 2).

     Separating cloud brightness values as a function of latitude is also helpful in determining the type of clouds formed in each region (Oreopoulos et al. 2014), although this is not included in our study. Studying these TOA reflectance value

distributions for opaque cloud cover will be helpful to understand shortwave albedo at regional and global scales. For example, opaque clouds compose a smaller percentage of global cloud cover with respect to transparent or spatially heterogeneous clouds (e.g., Rossow and Schiffer 1999; Stubenrauch et al. 2017) but deep convective clouds have an outsized influence on Earth's TOA radiative budget and hydrological cycle in the tropical latitudes (e.g., Jakob and Tselioudis 2003; Tan et al. 2015).

**3.3 Empirical Error Tests**

This section evaluates the stability of the threshold estimation approach and then quantifies cloud excision performance. Based on parameters such as latitude and surface type, we determined the false positive and false negative performance for different observing conditions. Screening classification depended largely on the false positive parameter (Fig. 5).

     It was critical that any threshold set generalized to new scenes not yet seen. While an infinitely large dataset would be sure to capture the real statistics of the globe, we were necessarily limited in the number of scenes used in the analysis. To

confirm this dataset was sufficiently representative to produce general thresholds, we performed two validation tests. First, we performed a leave-one-out cross-validation experiment, recalculating cloud-screening thresholds 102 times and excluding a different scene for validation from each trial. Every test conducted resulted in the same thresholds presented in Table 2. This stability is one validation that the optimal margin for the distribution for the brightness resolution of our lookup table was not sensitive to "outlier" cases but rather responded to the true statistics of global clouds. More generally, it confirmed that our

dataset was sufficient in space, time and wavelength to predict optimal thresholds for future scenes.

     In the second test, we conducted an experiment to measure the variance of the cloud-screening threshold estimates using bootstrap techniques. The 102 images used in the initial experiment were used to create a sample set through sampling



with replacement 500 unique times. Data concerning the threshold calculation were recorded each of the 500 times. Of particular interest is the variance in threshold calculation for each latitudinal zone (Table 5). The variance in each latitudinal

zone was low, showing that our estimation approach was robust.

## 3.4 Potential Improvement Yield of Screening

To evaluate the utility of cloud screening in terms of imaging spectroscopy science investigations, we analyzed the improvement yield for two specific cases of future orbital imaging spectrometers. We simulated the data return with and without the use of onboard cloud screening.

225        Our first case study used the orbital parameters of the Earth Surface Mineral Dust Source InvesTigation (EMIT) mission (Green et al., 2018). EMIT will use a visible to short wavelength infrared (VSWIR) imaging spectrometer to map the mineralogy of mineral-dust forming regions worldwide. This will improve our understanding of the mineral composition of airborne dust particles, informing the Earth System Models that simulate the dust cycle. Understanding the composition of mineral dust in Earth's atmosphere will in turn provide insight into the impact of dust on direct radiative forcing in Earth's

climate. The EMIT mission will be launched to the International Space Station in 2021, with an orbit dominated by low-latitude regions. A case study for EMIT provides one example of the potential for onboard cloud-screening (Table 6). Since the EMIT mission plan does not entail collecting data over the latitudinal zone specified as Ocean, we ignored that region in the case study.

       We intersected the EMIT coverage area with historical cloud probability maps to assess the potential improvement

for the instrument (Yelamanchili et al., 2019, Chien et al., 2019). We simulated cloud cover fractions using pre-calculated global cloud probabilities (x) from historical MODIS data (Mercury et al., 2012), defined as an annual average cloud cover probability at a spatial resolution of one degree. Then we simulated ISS observations at a 10 second rate for one year, starting on February 1st, 2022. The large improvement indicated the value of screening cloud-contaminated data. For the cloud screening approach in this work, we predicted at least double the current return of useful data in each latitudinal zone, and for

the mission overall. There is a notable difference in yield when considering all regions at once and when considering one region at a time. Some areas of the globe are cloudier than others, so the benefit of using a cloud-screening tool is particular to the region(s) of interest and the sampling strategy of each mission.

       As previously discussed, past literature presents cloud cover fractions that are greater over land than water and that tropical regions are more cloudy than other latitudinal zones (Eastman et al., 2012, Ackerman et al., 1998). The EMIT case

study cloud fractions showed that cloud cover in the tropics was greater than any other region studied. Due to the nature of our latitudinal zone sorting, we could not directly compare the Ocean cloud cover fractions with land cloud cover fractions; the other latitudinal zones could also have contained regions associated with Oceans.

       CHIME (Copernicus Hyperspectral Imaging Mission for the Environment) is an ESA (European Space Agency) mission that aims to provide routine hyperspectral observations of the Earth to aid in management of natural resources, assets

and benefits for the European Union. One main aspect of CHIME's technological advances is to facilitate increased field of



view (>5 deg) observations and low spatial sampling (< 30 m) with a high data rate (>1 Gbit/s) processor onboard the instrument. In light of this requirement, an onboard processing architecture such as a cloud screening algorithm is valuable to the CHIME mission.

The CHIME team conducted a simulated study, comparable to EMIT, where they reported simulated cloud cover based on one orbital cycle (223 orbits over 15 days) for summer 2012 with meteorological statistical inputs for cloud coverage simulation. The preliminary results of this study revealed that with a cloud screening algorithm onboard, CHIME would experience about a 50% increase in useable data (Table 6). In summary, both of the case studies considered facilitated at least double the return of viable data for each mission's desired objective.

     Another advantage to onboard cloud detection is the ability to, in turn, collect spectra of *only* clouds for studies
concerning cloud physics or instrument data products such as a stray light correction. Just as the process described above has the ability to recognize a cloud contaminated pixel and discard it from the data set, it can also set aside the cloud contaminated pixel for future use such as those described above.

## 4 Discussion and Conclusion

The method described for screening cloud-contaminated data onboard orbital imaging spectrometers will at least double the
volume of useful data for a fixed downlink size. We collected and studied a globally representative data set, producing optimal screening thresholds based on latitudinal zones. Using latitude as a parameter in screening clouds will help correctly classify cloud-contaminated pixels while reducing misclassifications of other surface types. The overall yield of useful data doubles when using the screening algorithm. In addition, we have produced a representation of cloud brightness in the 447.17 nm, 1249.36 nm, and 1650.90 nm wavelength bands, changing with latitude and surface type, based on TOA reflectance values.

Given the relatively low complexity and risk involved in implementing onboard cloud screening, it is a valuable option available to mission designers trying to achieve higher yield at low cost. It can be incorporated directly into instrument hardware, saving the storage costs of downlinking cloudy data. Its operation can be tuned over time, if needed, to obtain optimal performance for the specific observing profile of the mission. Even a very conservative threshold, with almost no probability of excising good data, can significantly reduce data volumes (Thompson et al., 2014). For this reason, the EMIT
mission (Green et al., 2018) will use this cloud screening method to optimize its downlink within the allowable resources provided by its position on the International Space Station. Its observation plan, which targets arid regions in the midlatitude and tropical latitudes, can use more aggressive thresholds without risking error due to snow cover. The data accumulated over its year-long mission will provide independent validations on these thresholds and brightness distributions.

     The results of this work also have applications outside screening clouds onboard imaging spectrometers. Conversely,
collecting solely cloud contaminated data that is identified using the algorithm described in this paper can aid in an instrument's stray light correction or other data products. On the other hand, the global study of cloud brightness shown in this paper has



the potential to be used in various cloud studies, for example those concerning cloud type and subsequent brightness as a function of latitudinal zone (Oreopoulos et al. 2014).

**Acknowledgements**

A portion of this research was performed at the Jet Propulsion Laboratory, California Institute of Technology (JPL). We acknowledge the support of the National Aeronautics and Space Administration (NASA). The NASA JPL summer internship program and the Imaging Spectroscopy group at JPL supported this work. Amruta Yelamachili and her colleagues at JPL helped to support the EMIT case study. Raffaele Vitulli and his colleagues at ESA helped to support the CHIME case study. Copyright 2020. All Rights Reserved.

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



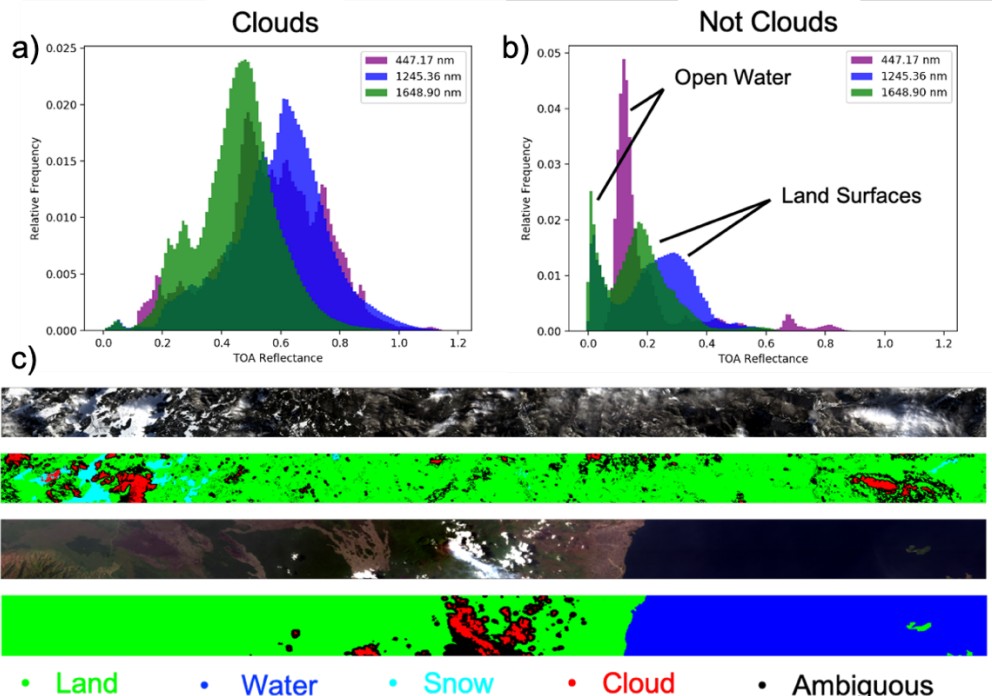

**Figure 1: An example of the one-dimensional distributions of a) cloudy and b) non-cloudy pixels in each wavelength created from c) the hand-labeled pixels in the Hyperion images used as ground truth.**

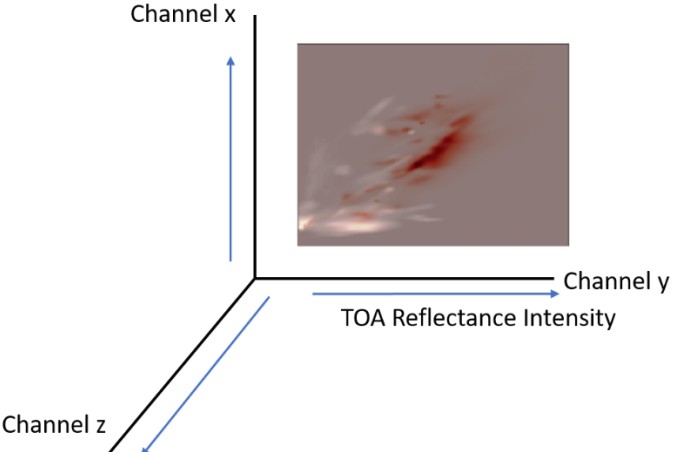

**Figure 2: A schematic of the three-dimensional cloud and non-cloud brightness distributions, with an example of a marginal distribution in one plane.**




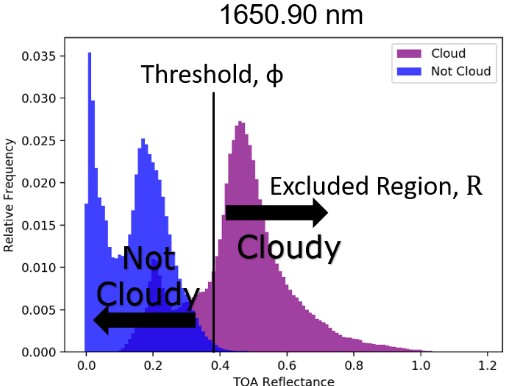

**Figure 3: Depiction of the exclusion region used to classify cloud-contaminated data.**

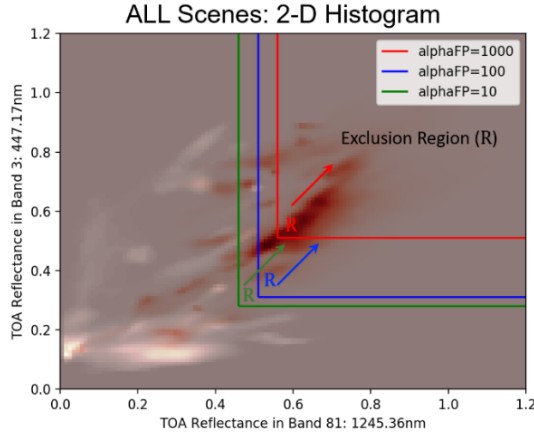

**Figure 4: A Two-dimensional histogram of cloud and non-cloud distributions for the Hyperion subset. The color gradient indicates relative frequency. The exclusion region for various false positives are shown as colored rectangles; αFP=1000 (red), αFP=100 (blue), αFP=10 (green).**

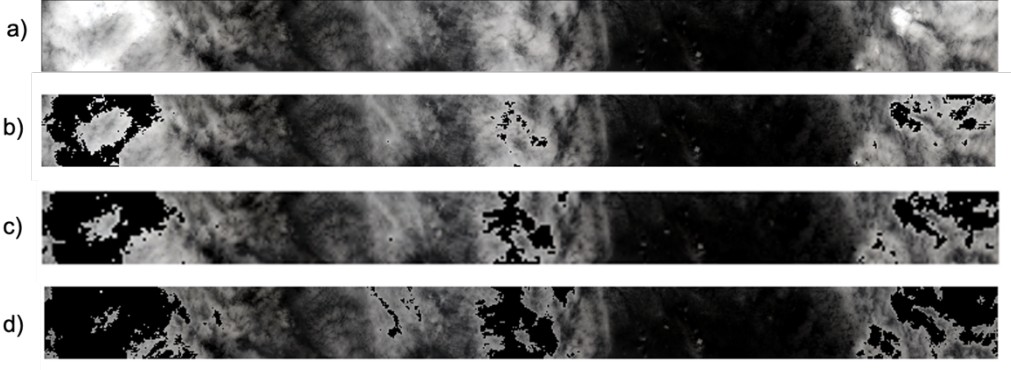


**Figure 5: An example of pixels classified as cloudy being excised from the image; a) the raw image with no screening, b) cloud-screening with αFP=1000, c) cloud-screening with αFP=100, and d) cloud-screening with αFP=10.**



**Table 1: The number of images in the Hyperion sample set (102 images) collected in each latitudinal zone. The ocean category also includes images taken in the regions of tropics, midlatitudes, and arctic.**

Hyperion Image Sample Set Breakdown

| Latitudinal Zone | Number of Images |
| --- | --- |
| Tropics | 30 |
| Midlatitudes | 51 |
| Arctic | 21 |
| ALL | 102 |
| Ocean Subset | 19 |


**Table 2: Optimal Thresholds in TOA (Top of Atmosphere) reflectance values using three difference false positives and a false negative of 1 for each latitudinal zone.**

Optimal Thresholds for Screening Clouds

| False Positive Value | Latitudinal Zone | 447.17 nm | 1245.90 nm | 1649.36 nm |
| --- | --- | --- | --- | --- |
|  | Tropics | 0.31 | 0.34 | 0.13 |
|  | Midlatitudes | 0.52 | 0.36 | 0.24 |
| 1000 | Arctic | 0.47 | 0.57 | 0.30 |
|  | Ocean | 0.41 | 0.37 | 0.30 |
|  | **ALL** | **0.51** | **0.56** | **0.29** |
|  | Tropics | 0.27 | 0.25 | 0.13 |
|  | Midlatitudes | 0.31 | 0.51 | 0.23 |
| 100 | Arctic | 0.55 | 0.27 | 0.22 |
|  | Ocean | 0.39 | 0.34 | 0.28 |
|  | **ALL** | **0.31** | **0.51** | **0.22** |
|  | Tropics | 0.26 | 0.21 | 0.11 |
|  | Midlatitudes | 0.28 | 0.45 | 0.22 |
| 10 | Arctic | 0.54 | 0.26 | 0.20 |
|  | Ocean | 0.32 | 0.25 | 0.22 |
|  | **ALL** | **0.28** | **0.46** | **0.22** |

**Table 3: Mean values of the distributions of TOA reflectance for each classification type, in each wavelength. The change in values**
**across this table verifies the advantage of presenting screening thresholds as a function of latitude.**





| Cloud Brightness mean values (TOA reflectance) | | | |
|---|---|---|---|
| Latitudinal Zone | 447.17 nm | 1245.36 nm | 1650.90 nm |
| Tropics | 0.47 | 0.61 | 0.50 |
| Midlatitudes | 0.57 | 0.61 | 0.45 |
| Arctic | 0.75 | 0.47 | 0.45 |
| Ocean | 0.49 | 0.49 | 0.44 |
| ALL | 0.49 | 0.61 | 0.48 |

**Table 4: Difference in mean values of the TOA reflectance thresholds in each zone from the overall thresholds. These differences are calculated by using the mean value in each band for the ALL category and subtracting the mean of each zone individually from this value.**

| Difference in Mean for Subset Latitudinal Zones vs ALL | | | |
|---|---|---|---|
| Latitudinal Zone | 447.17 nm | 1245.36 nm | 1650.90 nm |
| Tropics | 0.02 | 0 | -0.02 |
| Midlatitudes | -0.08 | 0 | 0.03 |
| Arctic | -0.26 | 0.14 | 0.03 |
| Ocean | 0 | 0.12 | 0.04 |


**Table 5: Variance in thresholds used to screen clouds based on bootstrapping the thresholds calculations described in the initial experiment 500 times. The variance is presented as a percentage of the mean of the particular zone. The false positive setting was $\alpha FP=1000$ and the false negative $\alpha FN=1$.**

| Variance in Thresholds from Bootstrapping (% of Mean) | | | |
|---|---|---|---|
| Latitudinal Zone | 447.17 nm | 1245.36 nm | 1650.90 nm |
| ALL | 1.63% | 1.77% | 0.58% |
| Tropics | 0.30% | 1.82% | 1.28% |
| Arctic | 1.08% | 2.74% | 1.96% |
| Midlatitudes | 1.29% | 0.87% | 0.52% |
| Ocean | 0.60% | 1.26% | 0.89% |

**Table 6: A case study of EMIT (Earth Surface Mineral Dust Source Investigation) concerning a global cloud fraction simulation was used to determine the improvement yield of the cloud-screening tool in terms of latitude.**

| Improvement Yield based on Case Studies | | |
|---|---|---|
| Case Study | Simulated Cloud Coverage Observed (%) | Improvement Yield (Factor of increase in usable data) |



| | | |
|---|---|---|
| EMIT | Tropics: 65% | x2.38 |
| | Midlatitudes: 57% | x2.85 |
| | Arctic: 52% | x2.32 |
| | Antarctic: 50% | x2.08 |
| | **All Zones: 58%** | **x2.04** |
| CHIME | **All Zones: ~ 50%** | **x2.00** |