# Peer review of "Global Cloud Property Models for Real Time Triage Onboard Visible-Shortwave Infrared Spectrometers"

_Atmospheric Measurement Techniques, 2020_

## Referee Comment (RC1) · Anonymous Referee #1 · 17 Jun 2020

**General Comments**

This manuscript presents a simple decision tree that delineates cloudy from cloud-free pixels onboard satellite-borne multispectral imagers. The filter allows to exclude unwanted pixels (in this case cloudy portions) before downlinking measurements to Earth and this enables transmitting about twice the rate of preferred pixels.

The document is well written and of interest to the community. I listed minor comments below that mainly aim at providing additional information and sharpening the line between "Results" and "Discussion". After resolving these comments, I recommend this manuscript for publication.

[Figure]

**Minor Comments**

l. 87: The reader could wonder where these labels are coming from. The Authors could put a brief link to Sec. 2.2 in anticipation of this question.

Sec. 2.1: Please list the pixel size and the orbit of the satellite. Can sun-glint be expected?

l. 100: Please briefly mention the selection process of 102 sample maps. How important is coverage across solar geometries versus surface types?

ll. 106-109: The purpose of this sentence is not apparent. Please rephrase or exclude if irrelevant.

ll. 117-119: I understand the satellite hardware limits the complexity of a cloud screening algorithm. How much more complex (than decision trees) could a potential algorithm be? What exactly are the limitation: RAM or CPU power? Perhaps these answers could extent the discussion in Sec. 4.

ll. 145-146: Should the 'historical average' be determined from the same pixel size as the future samples? (or in other words: does cloud fraction change when using larger or smaller pixels?) And what cloud optical thickness threshold was used for MODIS cloud detection? Perhaps the Authors could discuss these answers in the Sec. 4.

ll. 193-203: Perhaps these paragraphs are better suited for the discussion in Sec. 4.

ll. 259-262: Perhaps this paragraph is better suited for the discussion in Sec. 4.

Fig. 2: Please explain the colors in this figure.

Fig. 5: Which wavelength was used to capture this image? Please add the fraction of excluded pixels in b, c, and d to the caption.

[Figure]

---

## Referee Comment (RC2) · Anonymous Referee #2 · 30 Jun 2020

General comments:

This manuscript introduces a new onboard data analyzing method to increase the value of downlinked data by screening out cloud-contaminated data. An obvious advantage of the present work is that the thresholds used are surface-type dependent. The algorithm and results are tested to be robust and promising. Therefore, I recommend that this manuscript could be accepted once the authors could make some minor changes and provide more details (see specific comments).

Specific comments:

1. Section 2.1 (page 4): what is the pixel size of the Hyperion instrument? How do you

consider the instrumental differences between Hyperion and EMIT instrument (such as pixel size, wavelength, etc.)?

2. Section 2.2 (page 4): it is unclear to me how to manually label the 102 Hyperion images (7.7km x 42km)? Could you please provide more details?

3. Figure 2 (and also Figure 4): Could you please explain the meaning of colors and give a color bar on the side?

4. One of my major concern is that since this work could be potentially used in the EMIT mission, the authors should also consider the impact of dust aerosols. With the three channels selected in this study, it is possible that heavy dust cases were detected as clouds. I strongly suggest the authors use a case study to demonstrate the defined thresholds are also good for aerosols, in particular dust plumes.

5. Please consider cite a recent publication, which developed surface-type based machine learning models for cloud masking and cloud phase classification.

Wang, C., Platnick, S., Meyer, K., Zhang, Z., and Zhou, Y.: A machine-learning-based cloud detection and thermodynamic-phase classification algorithm using passive spectral observations, Atmos. Meas. Tech., 13, 2257–2277, https://doi.org/10.5194/amt-13-2257-2020, 2020.

---

## Author Comment (AC1) · 8 Aug 2020

macey\_sandford@outlook.com

Received and published: 8 August 2020

Response to RC1:

We would like to thank the reviewer for closely reading the article and providing helpful feedback while finding aspects of the discussion that can be expanded further on to clarify our work.

I. 87: The reader could wonder where these labels are coming from. The Authors could put a brief link to Sec. 2.2 in anticipation of this question.

Great idea! I can add that in.

Sec. 2.1: Please list the pixel size and the orbit of the satellite. Can sun-glint be expected?

The pixel size of the Hyperion instrument is 30 m per pixel and 7.5 km by 100 km land area per image. It followed a polar orbit.

I. 100: Please briefly mention the selection process of 102 sample maps. How important is coverage across solar geometries versus surface types?

To ensure that we sampled the entire globe, we collected approximately 25 images from each section of the globe. Namely the Arctic, Northern Midlatitude, Tropics, Southern Midlatitude, and Antarctic. We found that there were many less applicable images from both of the polar regions so combining them we included 21 images. We found that there were many Tropic images so that sample set ending up being 30 images and the Midlatitudes add up to 51. These number are not round because of the issue of resampling when including the Ocean subset. The 19 in the ocean category can be found in the other latitudinal zones, so any cross over between the two subsets were eliminated to ensure the entire sample set was unique.

Since we worked with TOA brightness, we did not need to consider solar geometries. According to our study of cloud brightness by latitude, surface type does play a role in apparent cloud brightness so it was important for us to include a variety of latitudinal ranges.

II. 106-109: The purpose of this sentence is not apparent. Please rephrase or exclude if irrelevant.

The purpose of these sentences is to inform readers about the selection of three bands to classify cloudy pixels. It gives background to the radiation sources that the instrument measures. These sentences will be written as: In order to detect cloudy pixels with confidence, we selected three specific spectral bands that can distinguish clouds from other surface types. It is important to note that Earth's total TOA energy flux con-
stitutes the total incoming solar radiation, the consequential outgoing reflected shortwave radiation from the clouds and surface, and the outgoing emitted longwave radiation from Earth's surface, atmosphere, and clouds (e.g., Trenberth et al. 2009).

II. 117-119: I understand the satellite hardware limits the complexity of a cloud screening algorithm. How much more complex (than decision trees) could a potential algorithm be? What exactly are the limitation: RAM or CPU power? Perhaps these answers could extent the discussion in Sec. 4.

The algorithm that we propose uses pre-calculated thresholds to screen data onboard. This method is of similar complexity to decision trees such that decision trees have N thresholds, where N is the maximum height of the tree, and our process uses 3 thresholds.

Hyperspectral instruments produce data rates of Gb/s and with relatively simple hardware (FPGA), or with hardware that people are starting to fly now (see below) you could keep up. Current efforts to fly more powerful computation include flight of the Qualcomm Snapdragon on the Mars Helicopter [Grip et al. 2019] and flight of the Intel Myriad Chip on FSSCCAT [ESA 2020], however future computing needs for onboard Al will continue to grow [Dally et al. 2020].

Dally, William J., Yatish Turakhia, and Song Han. "Domain-specific hardware accelerators." Communications of the ACM 63.7 (2020): 48-57.

Grip HF, Lam J, Bayard DS, Conway DT, Singh G, Brockers R, Delaune JH, Matthies LH, Malpica C, Brown TL, Jain A. Flight Control System for NASA's Mars Helicopter. In AIAA Scitech 2019 Forum 2019 (p. 1289).

European Space Agency (ESA), FSSCCAT-1 Ready for Launch, https://www.esa.int/Applications/Observing\_the\_Earth/Ph-sat/FSSCat\_F-sat-1\_ready\_for\_launch, retrieved 6 August 2020

For example, EO-1 had a Mongoose M5 which is a variant of a RAD 3000 (PowerPC
family) CPU. It had a 6 MHz clock speed so around 6 MIPS processing power. The challenge is that it had no hardware floating point support. If the computation was all performed fixed point it would be much more efficient, but all onboard classification was performed on top of the atmosphere reflectance data which was in floating point. Additionally, it had most but not all of the CPU for image processing. Therefore scenes required 10's of minutes to load into RAM and process, during which time another scene could not be acquired (as the Solid State Recorder could not simultaneously read and write). Current spacecraft have more computing capability but still fall far below laptop like computing power. A typical flight CPU would be a Rad 750 (about 200 MIPS) with 128 MB RAM. In comparison a typical laptop in 2020 has 400K MIPS or 2000x the compute power and 16 GB RAM. Future spacecraft are likely to have special purpose processors to handle instrument processing that would enable more sophisticated processing onboard. For example, the Mars 2020 Helicopter uses a Qualcomm Snapdragon processor.

II. 145-146: Should the 'historical average' be determined from the same pixel size as the future samples? (or in other words: does cloud fraction change when using larger or smaller pixels?) And what cloud optical thickness threshold was used for MODIS cloud detection? Perhaps the Authors could discuss these answers in the Sec. 4.

We actually did not experiment with how our historical averages change with pixel size. The best I can do to address this concern is point the reviewers to the method we used to label our ground truth pixels. The pixels surrounding areas of a certain surface type were labeled as ambiguous and not used in our cloud fractions so that any pixels that may include two classification types were not misclassified either way, thus not included in the historical average. This method helps to mitigate any change in cloud fraction where pixel size varies, for example if more than one surface type is present due to the large pixel size. Although, we could complete a study using data with smaller and larger pixels to definitively say that the pixel size is not a large concern, but it fell outside the scope of the study we conducted.
II. 193-203: Perhaps these paragraphs are better suited for the discussion in Sec. 4.

Thank you for the suggestion, I agree!

II. 259-262: Perhaps this paragraph is better suited for the discussion in Sec. 4.

Thank you for the suggestion, I agree!

Fig. 2: Please explain the colors in this figure.

Thank you for the suggestion, that would be helpful!

Fig. 5: Which wavelength was used to capture this image? Please add the fraction of excluded pixels in b, c, and d to the caption.

This Hyperion image is named EO1H1940712011304110T1. The image was created using the default RGB wavelengths in the ENVI program, although the algorithm to create the image in this figure only considers the three wavelengths used in our thresholds. The RGB image is used for visualization. The images in b, c and d show the pixels excised based on our calculated thresholds using a false positive of 1000, 100 and 10, respectively. Figure 5b has 8.5% of the total pixels excised, figure 5c has 16.4%, and figure 5d has 25%. I plan to revise the figure caption to include the filename and the percentage of excised pixels.

AMTD

---

## Author Comment (AC2) · 8 Aug 2020

Response to RC2:

We appreciate the feedback from this reviewer and have found the comments to be helpful in clarifying our work in this article.

1. Section 2.1 (page 4): what is the pixel size of the Hyperion instrument? How do you C1 AMTD Interactive comment Printer-friendly version Discussion paper consider the instrumental differences between Hyperion and EMIT instrument (such as pixel size, wavelength, etc.)?

[Figure]

The pixel size of the Hyperion instrument is 30 m per pixel and 7.5 km by 100 km land area per image. EMIT has a spectral resolution from 380-2510 nm and a spatial resolution of 30 m per pixel, both very similar to Hyperion.

2. Section 2.2 (page 4): it is unclear to me how to manually label the 102 Hyperion images (7.7km x 42km)? Could you please provide more details?

We manually labeled the pixels in each scene using an image editing software called GIMP. In this software we manually labeled pixels by human classification, visually. Depending on the classification, the pixel was given a color (in value); red, green, blue, cyan, or black. Black pixels bordered each other classification type to mitigate misclassifications or ambiguous areas. It took a long time, but having a ground truth classification of surface type will be helpful to studies past our own.

3. Figure 2 (and also Figure 4): Could you please explain the meaning of colors and give a color bar on the side?

The red color is clouds and the white color is non-clouds. Yes, this should be included in both figures.

4. One of my major concern is that since this work could be potentially used in the EMIT mission, the authors should also consider the impact of dust aerosols. With the three channels selected in this study, it is possible that heavy dust cases were detected as clouds. I strongly suggest the authors use a case study to demonstrate the defined thresholds are also good for aerosols, in particular dust plumes.

This is an astute point and warrants some additional discussion of the topic in the conclusion. Fortunately EMIT is somewhat immune to this problem because EMIT will not actually measure mineral dust in the atmosphere. It is a geologic mapping mission to map the mineralogy of mineral dust source areas. In fact, the mission intends to filter any AOD550 higher than 0.4. From this perspective, it is fine if the dust plume is screened, because that data would not have been used anyway. Also, it is not nec-

essary that the cloud screening method detect such plumes, because the mission has other methods for estimating AOD550 in the Level 2 stage. By filtering obviously obscured scenes, cloud screening will reduce transmitted data volumes by approximately 50%, enabling EMIT to achieve its geologic mapping objectives in the first 6 months of operation.

5. Please consider cite a recent publication, which developed surface-type based machine learning models for cloud masking and cloud phase classification.

Wang, C., Platnick, S., Meyer, K., Zhang, Z., and Zhou, Y.: A machine-learning-based cloud detection and thermodynamic-phase classification algorithm using passive spectral observations, Atmos. Meas. Tech., 13, 2257–2277, https://doi.org/10.5194/amt-13-2257-2020, 2020.

Great addition! Thank you.

———————————————————